# Application-Aware Scheduling for IEEE 802.15.4e Time-Slotted Channel Hopping Using Software-Defined Wireless Sensor Network Slicing

**DOI:** 10.3390/s23167143

**Published:** 2023-08-12

**Authors:** Tarek Sayjari, Regina Melo Silveira, Cintia Borges Margi

**Affiliations:** Escola Politécnica, Universidade de São Paulo, São Paulo 05508010, Brazil; regina@larc.usp.br (R.M.S.); cintia@usp.br (C.B.M.)

**Keywords:** software-defined wireless sensor networks, time-slotted channel hopping, network slicing, quality of service, application traffic isolation

## Abstract

Given the improvements to network flexibility and programmability, software-defined wireless sensor networks (SDWSNs) have been paired with IEEE 802.15.4e time-slotted channel hopping (TSCH) to increase network efficiency through slicing. Nonetheless, ensuring the quality of service (QoS) level in a scalable SDWSN remains a significant difficulty. To solve this issue, we introduce the application-aware (AA) scheduling approach, which isolates different traffic types and adapts to QoS requirements dynamically. To the best of our knowledge, this approach is the first to support network scalability using shared timeslots without the use of additional hardware while maintaining the application’s QoS level. The AA approach is deeply evaluated compared with both the application traffic isolation (ATI) approach and the application’s QoS requirements using the IT-SDN framework and by varying the number of nodes up to 225. The evaluation process took into account up to four applications with varying QoS requirements in terms of delivery rate and delay. In comparison with the ATI approach, the proposed approach enhanced the delivery rate by up to 28% and decreased the delay by up to 57%. Furthermore, even with four applications running concurrently, the AA approach proved capable of meeting a 92% delivery rate requirement for up to 225 nodes and a 900 ms delay requirement for up to 144 nodes.

## 1. Introduction

Wireless sensor networks (WSNs) became increasingly adopted in a wide range of application domains, including security, environment, health, and military. These networks consist of sensor nodes with limited resources concerning energy, memory, and processing [1,2]. However, the development of WSN applications nowadays requires a high level of quality of service (QoS). In this context, software-defined networking (SDN) [3,4] is applied with WSNs to increase flexibility. Software-defined wireless sensor networks (SDWSNs) facilitate resource sharing and network reconfiguration [5]. They transfer all management and control tasks to a centralized controller, which has an external power supply. IT-SDN [6] is an SDWSN framework example that has been thoroughly analyzed [7], with the results demonstrating that the performance is comparable with that of the routing protocol for low-power and lossy networks (RPL) [8]. Yet, as well as other wireless networks, different traffic types compete for limited resources across the shared wireless medium, making it problematic to ensure the required QoS level [9].

The IEEE 802.15.4e time-slotted channel hopping (TSCH) [10,11] technology demonstrated its efficacy with the WSNs [12]. TSCH was evaluated [13] using various scheduling approaches, and the results showed that it was capable of improving the data-delivery rate and delay. TSCH was thus employed to address competition for available resources through slicing [9,14]. TSCH slices the network into slotframes, which are repeated throughout time. Each slotframe is a set of time instants called timeslots. The TSCH schedule specifies when network nodes must send and receive messages during these timeslots [10].

Several issues about SDWSN on top of TSCH were investigated, including interference and multipath fading [15], traffic competition over the wireless medium [9,14,16], mobility management [17], and ensuring the application’s requirements [18,19]. Each of these works concentrated on some scheduling approach to handle these issues. However, most of them did not support network scalability since they used dedicated timeslots in the scheduling approach. Orozco-Santos et al. [20] addressed the scalability issue over SDWSNs. Nonetheless, the proposed approach was based on the creation of virtual sinks, which necessitates the addition of several radio interfaces. The literature review then revealed that there is still a gap in providing the required QoS level for scalable SDWSNs without requiring any extra radio interfaces.

Since using shared timeslots allowed for isolating the different traffic types and supported the network scalability [9,16], we were encouraged to investigate the usage of these timeslots to maintain the required QoS level. The main contribution of this work is the design and evaluation of the application-aware (AA) scheduling approach, which dynamically alters scheduling to ensure the application’s QoS requirements in terms of delivery rate and delay. To the best of our knowledge, this is the first paper to address the issue of ensuring the application’s QoS requirements for scalable SDWSNs using shared timeslots. The AA approach is evaluated in two different strategies: (i) compared with the application traffic isolation (ATI) approach [16] and (ii) compared with the application’s QoS requirements. We adopt the IT-SDN [6] framework and up to four applications with different priorities (QoS requirements). Moreover, we considered a network with up to 225 nodes and 6 different simulation scenarios varying the metric calculation rate (MCR) and the data traffic rate (DTR). The simulations were carried out using Contiki OS [21] and COOJA [22] simulators. The results showed that the AA approach increased the delivery rate by up to 28% and decreased the delay by up to 57% in comparison with the ATI approach. Moreover, it ensured the application’s QoS requirements for an increasing number of nodes.

The remainder of the paper is structured as follows: Section 2 reviews the literature, and Section 3 presents a fundamental background about SDWSNs and TSCH. The AA approach is described in Section 4, and the research method is presented in Section 5. The performance evaluation is provided in Section 6, and the paper is concluded in Section 7.

## 2. Related Work

As aforementioned, TSCH proved its efficiency with the limited resources networks [12,23] in recent years. SDWSNs, thus, were adopted in several studies on top of TSCH to overcome challenges, such as interference and multipath fading [15], competition among the different traffic types [9,14,16], mobility management [17], ensuring the required QoS level [18,19], and the scalability issue [20]. This section highlights and reviews such works in the literature.

Thubert et al. [15] used 6TiSCH to build centralized scheduling managed via the path computation element (PCE) protocol [24]. The issues of multipath fading and interference were addressed, and the proposed architecture was theoretically presented with no implementation or evaluation methods.

Baddeley et al. [14] constructed dedicated tracks to accommodate the control messages and to isolate them from the data ones. Each track is a path consisting of a set of nodes between a node and the controller entity. As for the SDWSN framework, the authors adopted μSDN, which includes only two control message types. In addition to the tracks, the scheduling approach assigns four shared timeslots for the data messages. To evaluate the efficiency of the proposed approach, a comparison process was carried out between the adopted scheduling with and without the tracks. The simulation results showed that the end-to-end delay of the application layer was reduced in the case of the TSCH tracks.

Lo Bello et al. [17] proposed forwarding and TSCH scheduling over SDN (FTS-SDN) to address the topology changes caused by the node mobility. The authors adopted SDN-WISE [25], and the proposed approach uses the 6TiSCH minimal scheduling [12] during the network setup period. After that, two or more timeslots are assigned to each node; one of them is assigned to accommodate the broadcast messages, and the others are assigned to the unicast messages to dynamically adapt to the topology changes. To evaluate the proposed approach, a single mobile node moving at three different speeds is considered. The results showed that the end-to-end data delay is lower when the FTS-SDN approach is adopted.

Orozco-Santos et al. [18] improved the SDN-WISE framework by adopting TSCH technology as the media access control (MAC) layer. They aimed to dynamically ensure the application requirements in terms of packet loss rate and time delay. Different priorities were assigned to the considered applications depending on the application’s requirements. The authors added three modules to the application layer, namely traffic manager, routing process, and TSCH scheduler. These modules work with the controller to dynamically decide the adequate route and schedule. All implementation details were provided, and the evaluation process was achieved using simulation and Testbed. The results showed that the proposed approach increased the network lifetime and ensured the application’s requirements.

Sayjari et al. [9] isolated the data messages from the control ones using shared timeslots. One timeslot was assigned to accommodate the control messages, in addition to several timeslots for the data ones. 6TiSCH minimal scheduling [12] was adopted as the reference comparison case. The proposed approach was evaluated considering control and data planes, and the simulation results showed that it improved the packet delivery rate and delay in both planes.

Orozco-Santos et al. [26] investigated the importance of SDN on top of TSCH. They compared several TSCH schedulers, namely SDN WISE-TSCH [18], the adaptive multi-hop scheduling method (AMUS) [27], the 6TiSCH minimal scheduling function (MSF) [28], and Orchestra [29]. All these schedulers were considered in the first stage of the simulation. Then, the two schedulers that presented the best performance in the simulation (SDN WISE-TSCH and Orchestra) were selected to be compared using Testbed. The results showed that SDN WISE-TSCH outperformed the other schedulers.

Veisi et al. [19] proposed the SDN-TSCH approach. They aimed to control the reliability and delay values of the considered flows. The controller entity defines the adequate schedule and reserves the required resources for each flow. SDN-TSCH also adopts the concept of control and data traffic isolation and used dedicated timeslots to construct reliable paths from the nodes to the controller and vice versa. Each traffic flow is also accommodated using dedicated timeslots. The authors considered up to 15 nodes and compared their approach with Orchestra [29]. The simulation results showed that the proposed approach improved the delivery rate and delay in comparison with Orchestra, especially for the network size of 15 nodes.

Orozco-Santos et al. [20] addressed the scalability issue in the industrial WSNs (IWSNs). They considered that the sink is the only node that connects the controller with the SD-IWSN. Thus, the sink suffers from congestion. This, in turn, limits the number of nodes. The proposed approach used the advantages of SDN to add a virtual sink, which allows for network extension. Normally, each node has a single radio interface. This means that the sink node (as well as the other nodes) can receive from only one transmitter node during a timeslot. The virtual sink could be considered as a set of nodes, each one with a radio interface, making the controller consider them as a single sink. In other words, this work considered sinks with several radio interfaces to extend the capabilities of the real sink. A sink node with three radio interfaces, for instance, is capable of receiving from three different nodes at the same timeslot. The results showed that virtual sinks allowed for network extension and helped to meet the application’s requirements.

Sayjari et al. [16] aimed to investigate the effect of application’s traffic isolation on network performance. The proposed application traffic isolation (ATI) approach also adopts the concept of control and data traffic isolation. The ATI approach uses the shared timeslots and assigns a single timeslot per application traffic. The reference case for comparison was the control and data traffic isolation (CDTI) approach [9], and the results showed that the ATI approach increased the delivery rate and reduced the delay in the application layer.

Table 1 represents a comparison among the related works. It confirms that the present work is the first to consider the application requirements and supports the network scalability using shared timeslots.

## 3. SDWSNs and IEEE 802.15.4e TSCH

We consider an architecture based on SDWSN over TSCH, and this section presents the main concepts of these technologies.

### 3.1. Software-Defined Wireless Sensor Networks (SDWSNs)

SDWSNs result from the application of the SDN paradigm to the WSNs. These networks enable resource sharing and reuse and improve network management and configuration. They decouple the control plane from the data one, and all the management and control decisions are made using a centralized controller, in which the sensor nodes become forwarding devices [5]. The SDWSN architecture is divided into three planes: (i) the infrastructure plane, which includes the sensor nodes (that sense the environment and send data messages to the sink (s)) and communicates with the control plane using the southbound (SB) protocol; (ii) the control plane, which achieves all control and management tasks through a centralized controller; and (iii) the application plane, which communicates with the control plane using the northbound (NB) protocol.

We adopt IT-SDN as the SDWSN framework. The IT-SDN uses three separated protocols: (i) the SB protocol, which ensures communication between the SDN nodes and the controller; (ii) the neighborhood discovery (ND) protocol, which obtains information about SDN node neighbors; and (iii) the controller discovery (CD) protocol, which specifies the next hop on the path between the current node and the controller. The SB protocol comprises the following packet types: flow request, flow setup, flowid register, neighbor report, acknowledgment, and data. The flow request packet requests the controller information about packets without entries in the flow table, and the controller replies using a flow setup packet with the associated configuration. The neighbor report packet carries the node neighborhood information, and the acknowledgment packet confirms the delivery of the control packets. All the routes are calculated with the controller according to the given rules. IT-SDN was thoroughly evaluated [7] by varying several parameters, and the results confirmed its efficiency in comparison with the routing protocol for low-power and lossy networks (RPL) [8].

### 3.2. IEEE 802.15.4e Time-Slotted Channel Hopping (TSCH)

This section introduces the time-slotted channel hopping (TSCH) technology, which serves as the MAC layer in the architecture we’ve chosen. The definition of TSCH is highlighted first, followed by the scheduling concept.

#### 3.2.1. Definition

IEEE 802.15.4e time-slotted channel hopping (TSCH) [10] was designed for low-power and lossy networks (LLNs) to provide a reliable MAC layer. It uses time-slotted access to divide the time into several timeslots, clustered into one (or more) slotframe(s). Both the effects of congestion and collision are reduced via TSCH using the channel hopping feature since different channels (frequencies) are used to achieve the transmit and receive processes. Each TSCH node sends enhanced beacon (EB) messages periodically. When a node wants to join a TSCH network, it must listen to an EB. Thus, it randomly selects a channel and listens to it for a period before selecting another channel and performing the same operation. The process continues until the node receives an EB message and associates it with the TSCH network.

#### 3.2.2. Scheduling

TSCH provides the concept of scheduling. Each pair of timeslot and channel offset is called a cell. There are three types of timeslots [10]: (i) shared, where all the transmit and receive processes are achieved; (ii) dedicated, where only a single process is achieved; and (iii) idle, where no process is achieved. The scheduling determines, for each node, when to transmit, receive, or sleep. It is considered, thus, one of the most important aspects concerning TSCH. The minimal mode of operation for TSCH is called 6TiSCH minimal scheduling [12], which consists of only one shared cell. Various scheduling approaches were proposed in the literature, and TSCH proved its efficiency with the limited resources of the WSNs [23].

## 4. Application-Aware (AA) Scheduling Approach

The main contribution of this work is the application-aware (AA) scheduling approach, which modifies the current scheduling approach in real time to ensure the application’s QoS requirements. Figure 1 shows the adopted system architecture, which consists of three planes: (i) the infrastructure plane, (ii) the control plane, and (iii) the application plane. The control plane contains the IT-SDN controller, which periodically receives the traffic data from the infrastructure plane, calculates the delivery rate and delay metrics, sends them to the application plane, and disseminates the new schedule to the network nodes. Two modules were added to the application plane: (1) the application manager module, which periodically receives the calculated metrics, checks if the application’s QoS requirements are met, and asks the TSCH scheduler to assign adequate scheduling, and (2) the TSCH scheduler module, which calculates the new scheduling and sends it to the controller.

To be aware of the application’s requirements, the different planes of the adopted system exchange five types of control messages:Statistical message: it counts the sent/received data messages, besides recording the message’s arrival time. It is periodically sent from the sensor nodes and sinks to the controller.Calculated metrics message: includes the calculated delivery rate and delay metrics for each of the considered applications. It is periodically sent from the controller to the application manager module.Rescheduling request message: contains the number of timeslots that should be added to/removed from each application. It is sent from the application manager module to the TSCH scheduler module.Rescheduling message: it is sent from the TSCH scheduler module to the controller and contains the new scheduling.New scheduling message: it is sent from the controller to the network nodes and contains the new scheduling.

We consider applications with four different priority levels, as Table 2 shows. The highest priority is assigned to the first application type, whereas the lowest priority is assigned to the last application type (type 4). The applications are classified depending on their delivery rate and delay requirements.

Since control and data traffic isolation and the search feature [16] improved the network performance, we adopt these concepts in our proposed approach. Moreover, to speed up the network convergence, the control messages are transmitted in all the timeslots during the convergence period. After this period, the data messages start to be transmitted using Sch 0 scheduling, as Figure 2 shows. The first timeslot(s) are always reserved for the control messages. The slotframe size is selected to accommodate the additional applications and to prioritize the applications with harder requirements. In Figure 2b, where two applications are considered, nine timeslots (56.25% of the slotframe size) are assigned to Application 1, compared with six timeslots (37.5% of the slotframe size) assigned to Application 2.

The application plane delivery rate and delay metrics are periodically calculated. Each sensor node periodically (according to the MCR value) sends to the controller the tuple (DS, TXt) that represents the number of the data messages sent (DSid) and the timestamp of the data message sent (TXt). Additionally, the sink periodically sends to the controller the tuple (DR, RXt), which similarly represents the number of data messages received (DRid) and the timestamp of the data message received (RXt). The data delivery rate and delay metrics are calculated using Equations (Equation 1) and (Equation 2).
(1)$$\begin{array}{cc} Delivery\;rate\;(\%) & =100\times \frac{DRid}{DSid} \end{array}$$
(2)$$\begin{array}{cc} Delay(s) & =\frac{\Sigma \;(RXt-TXt)}{DSid} \end{array}$$

Since the considered applications send data messages at different DTR values, the node does not send its statistical message about an application if the number of transmitted messages for this application is zero. To explain this, suppose that the DTR value of Application 2 is 10 min. This means that the first data message for this application is transmitted after 10 min from the start of the run-time. During these 10 min, the nodes do not send statistical messages for Application 2 to the controller. The calculated metrics are sent to the application manager module to verify if the application’s QoS requirements are satisfied. If so, Sch 0 continues to be the adopted scheduling. Otherwise, the TSCH scheduler module assigns a new scheduling and sends it to the controller, which in turn, disseminates it to the network nodes. Figure 3 shows the sequence of all the system operations.

Concerning the new scheduling calculation, more (or fewer) resources are assigned to ensure the application’s QoS requirements. The proposed approach adds timeslots to the end of the slotframe depending on the calculated metrics values, as follows:The calculated metric is (0–20)% worse than the application’s requirement; a single timeslot is added to the end of the slotframe.The calculated metric is (20–40)% worse than the application’s requirement; two timeslots are added to the end of the slotframe.The calculated metric is (40–60)% worse than the application’s requirement; three timeslots are added to the end of the slotframe.

To generalize the rule of adding more timeslots, we could say that if the calculated metric is from N% to (N + 20)% worse than the application’s requirement, X timeslots are added to the end of the slotframe. This leads to saying that if the calculated metric is from (N + 20)% to (N + 40)% worse than the application’s requirement, (X + 1) timeslots are added to the end of the slotframe. Similarly, when the calculated metric is better than the application’s requirement, the proposed approach removes timeslots from the end of the slotframe as follows:The calculated metric is (0–20)% better than the application’s requirement; no timeslots are removed.The calculated metric is (20–40)% better than the application’s requirement; a single timeslot is removed from the end of the slotframe.The calculated metric is (40–60)% better than the application’s requirement; two timeslots are removed from the end of the slotframe.

As a general rule, for N greater than or equal to 20, if the calculated metric is from N% to (N + 20)% better than the application’s requirement, X timeslots are removed from the end of the slotframe. This leads to saying that if the calculated metric is from (N + 20)% to (N + 40)% better than the application’s requirement, (X + 1) timeslots are removed from the end of the slotframe. Figure 4 shows the scheduling calculation procedure considering that MV is the metric value, AR is the application’s requirement value, and TS represents the timeslot. Assigning multiple timeslots per application at the same time (if needed) could help to reduce the time required to ensure the application’s requirements. Removing multiple timeslots (if needed), on the other hand, could help to save energy and increase the network lifetime.

If the two requirements of an application are not satisfied, the number of added timeslots will depend on the highest number of required timeslots for each requirement. To explain this, suppose that during a metric calculation cycle, the delivery rate of Application 1 was 10% lower than the requirement, and the delay was 30% higher than the requirement. The number of added timeslots for Application 1 would thus be 2 timeslots. Similarly, if the 2 requirements of Application 1 are better than the required values, the number of the removed timeslots will depend on the lowest number of required timeslots for each requirement. For instance, suppose that during a metric calculation cycle, the delivery rate of Application 1 was 10% higher than the requirement, and the delay was 30% lower than the requirement. The number of removed timeslots for Application 1 would be 1. To clarify how the new scheduling is constructed, suppose that during a metric calculation cycle, multiple applications were not satisfied; the timeslots would then be added depending on the application’s priorities. Suppose that in the case of three running applications, Applications 1 and 2 were not satisfied, and two more timeslots should be assigned to Application 1 and one more timeslot to Application 2. Then, the Sch 0 scheduling depicted in Figure 2c turns into the scheduling depicted in Figure 5, where firstly, two more timeslots were assigned to Application 1 (which has the highest priority), and then one more timeslot was assigned to Application 2.

Our approach is the first one to dynamically adapt to the application’s requirements for scalable SDWSNs without any additional hardware. Moreover, the Sch 0 schedule, which is a part of the AA approach, considers the application’s priorities, which could delay the need for adding more timeslots. This, in turn, could save both energy and the time required for rescheduling. The AA approach reduces the time required to ensure the application’s QoS requirements since the number of added timeslots depends on the difference between the calculated metrics and the application’s requirements. Our approach could also save on energy since it contains a mechanism to dynamically remove active timeslot (s) when possible. Finally, this approach assigns more shared timeslots per application if needed, without any dedicated timeslots, in which it is difficult to support large networks and more energy is consumed.

## 5. Method

We adopt IT-SDN [6] as the SDWSN framework, whereby the simulations were carried out using Contiki OS [21] and a COOJA simulator [22]. Each application is represented by a set of sensor nodes that periodically send data messages to a single sink. To execute up to 4 applications with different DTR values, each sensor node periodically sends a data message to all the sinks. Hence, each sensor executes up to 4 applications simultaneously. Both the IT-SDN controller and the TSCH coordinator (which controls join and departure operations within the TSCH network) run on the same node. We consider up to 4 application types with different DTR values and network sizes of up to 225 nodes. Each experiment is repeated 10 times, and its duration is set to be 1 h. The shown simulation results are the average of these repetitions.

To evaluate the AA approach, we compare it in two different strategies: (i) adopting the application traffic isolation (ATI) [16] approach as a reference case for comparison and (ii) comparing the application’s QoS requirements with the obtained metrics. For all the simulation results, we show the performance for each of the considered applications in both control and data planes. It is important to define two parameters related to the AA approach: (i) metric calculation rate (MCR), which is the frequency of metric’s (delivery rate and delay) calculations during the run-time, and (ii) difference rate, which is the percent of the difference between the calculated metrics and the application’s QoS requirements. Table 3 shows the default simulation settings.

To compare the ATI approach, which assigns a single timeslot per application, with the AA approach, we consider the case of 4 applications running simultaneously. Figure 6 depicts the simulation scenario for each approach.

To investigate the ability of the proposed approach to ensure the application’s QoS requirements, we compare the results with the application’s requirements. Six different scenarios were considered for this evaluation strategy, varying both MCR and DTR values. Table 4 shows the parameters’ values for these scenarios, indicating the default values with “DV”.

Performance is evaluated using the following metrics:Data delivery rate: the total number of received data messages divided by the total number of sent data messages;Data delay: the average time a data message takes to reach its destination;Control overhead: it includes IT-SDN control messages (flow request, flow setup, source routed flow setup, acknowledgment, neighbor discovery, controller discovery, and neighbor report messages), in addition to the AA approach control messages (statistical messages, calculated metrics messages, rescheduling request messages, rescheduling response messages, and new scheduling messages);Energy consumption: it is represented by the average energy consumed by the node;Control delivery rate: the total number of received control messages divided by the total number of sent control messages;Control delay: the average time a control message takes to reach its destination.

## 6. Results and Discussion

This section presents the simulation results, which are divided into (i) a comparison with the ATI approach and (ii) a comparison with the application’s QoS requirements. All of these results take into account both the control and data planes.

### 6.1. Comparison with the ATI Approach

In this section, we compare our proposed approach with the ATI approach. Figure 7 depicts the performance of both the AA and ATI approaches in the data plane for four concurrent applications.

Concerning the delivery rate and delay, Figure 7a,b indicate that for the first three applications (first application, second application, and third application), the AA approach outperforms the ATI approach. For a network size of 196 nodes and in the case of the second application, the delivery rate and delay of the AA approach are 90.84% and 1.4 s compared with 62.2% and 3.27 s, respectively, for the ATI approach. This occurs because the AA approach allocates priority and adapts to the application’s requirements dynamically. For the fourth application, the ATI approach outperformed the AA approach in terms of both delivery rate and delay since in the case of the AA approach, the fourth application has no priority and is accommodated using only one timeslot, as shown in Figure 6b.

Figure 7c shows that the control overhead of the AA approach is higher than that of the ATI approach regardless of the number of nodes because the AA approach employs more control message types to meet the application’s QoS requirements. For a network of 64 nodes, the control overhead value is 10,927.1 messages for the AA approach compared with 4036.7 messages for the ATI approach. The AA approach consumed more energy than the ATI approach, as seen in Figure 7d. In comparison with the ATI strategy, the AA approach increased energy usage by 14.3% and 16.27% for networks of 100 and 144 nodes, respectively. This is primarily due to the exchange of control messages. This mainly occurs due to the exchanged control messages. Since the ATI approach does not add any control messages, then the number of exchanged control messages after the network convergence is relatively low. This means that the nodes sleep for longer periods during the control timeslots. The nodes in the case of the AA approach, in turn, continue to exchange several control message types after the network convergence, and so should stay awake longer.

Concerning the control plane, Figure 8 depicts the control delivery rate and control delay for the case of four applications running simultaneously. Figure 8a,b show that, in general, both approaches present similar values for the increasing number of nodes, and the ATI approach performs slightly better than the AA approach. For a network size of 144 nodes, the ATI approach increased the control delivery rate by 4.65% and decreased the control delay by 7.45% in comparison with the AA approach. This could be justified by the number of exchanged control messages after the network convergence for both approaches. This number is higher for the AA approach, which increases the probability of packet drop due to the buffer fullness and increases the time that the message waits to be processed.

Table 5 summarizes the comparison between the AA and ATI approaches, highlighting the outperformed approach for each of the evaluation metrics.

### 6.2. Comparison with the Application’s QoS Requirements

This section provides a comprehensive assessment of the AA approach. We investigate the impact of MCR and DTR values on network performance. For each case, we consider up to four applications running concurrently.

#### 6.2.1. Metric Calculation Rate (MCR)

In this section, we investigate the impact of the MCR value considering various scenarios. It should be noted that the low values of MCR could speed up ensuring the application’s QoS requirements. This, however, results in a large control overhead. High values of MCR, conversely, may postpone ensuring the application’s QoS requirements, because the AA approach needs more time to discover the unsatisfied application. However, this reduces the control overhead.

Figure 9 depicts the data delivery rate for up to four applications, considering four different scenarios. For the case of a single application (Figure 9a), the delivery rate’s requirement is ensured for most of the considered scenarios and the number of nodes, except the case of Scen 4. This occurs since the MCR value is relatively high, and the metrics are calculated every 8 min during the run-time.

Concerning two applications, Figure 9b shows that the first application is always satisfied for most of the considered scenarios. However, for the network size of 225 nodes, the obtained delivery rate values are a little lower than the requirement for all the considered scenarios. The second application presents a delivery rate value not lower than 90% for most cases, although the second application does not have a delivery rate requirement. This is justified by the delay requirement of the second application, which induces the AA approach to be active. This, in turn, improves the delivery rate.

Figure 9c depicts the case of three applications. The delivery rate requirement of the first application of all scenarios and for up to 100 nodes is always satisfied. For a higher number of nodes, the delivery rate is a little lower than the requirement for the rest of the cases. This is justified by the probability of packet drop resulting from the high traffic of the three applications. For the third application, the delivery rate requirement is satisfied for all the numbers of nodes and scenarios. This occurs since, in addition to the efficiency of the AA approach, the data traffic rate of the third application is 8 min. Therefore, the probability of a packet drop resulting from the buffer fullness is low.

Concerning four applications running simultaneously as Figure 9d shows, we notice that increasing the MCR value resulted in a lower delivery rate, especially for network sizes larger than 144 nodes. For the first application, the sole scenario capable of ensuring the delivery rate requirement for all the network sizes is Scen 2, where the MCR value is 1 min. For the remaining cases, the AA approach was able to ensure the requirement for up to 144 network sizes. For the third application case, the AA approach ensured the requirement for all cases. This mainly occurs since the DTR value of the third application is low (one data message every 8 min). The fourth application presents the worst values for all scenarios since this application has no priority and is accommodated using only one timeslot.

Concerning the data delay, Figure 10a shows that in the case of a single application, the delay requirement, which is 900 ms, is ensured for network sizes up to 144 nodes for all scenarios. For larger networks, the obtained delay values are a little higher than 900 ms. For a network size of 196 nodes, the delay values were 0.94 ms and 0.97 ms for Scen 1 and Scen 3, respectively.

Figure 10b shows that for two applications, the delay requirement is ensured for network sizes up to 100 nodes, compared with the 144 nodes in the case of a single application. This happens because more applications mean more data messages. This increases, in turn, the time the message waits in the buffer to be processed. Concerning the second application, reducing the MCR value did not enhance the obtained values considerably. This is justified by the DTR value of the second application, which is 4 min. This, in turn, demonstrates a significant trade-off between the chosen MCR and DTR values.

Concerning the case of three applications running simultaneously depicted in Figure 10c, the delay requirement of the first application is ensured for network sizes of up to 100 nodes when the MCR value varies between 1 and 4 min. However, the AA approach ensures the delay requirement only for network sizes up to 64 nodes when MCR is higher (6 and 8 min). This occurs because the higher MCR values could delay ensuring the requirement. The requirement is unsatisfied for a longer period during the run-time, which affects the final delay result. For the third application, the requirement keeps ensuring up to network sizes of 100 nodes for all scenarios, except for Scen 4, where the MCR value is 8 min.

For four applications, as depicted in Figure 10d, and for the first application, the delay values are lower than 900 ms for up to 144 nodes in Scen 1 and Scen 2, and up 100 nodes in the Scen 3 and Scen 4, where the MCR values are 6 and 8 min, respectively. For the second application, the delay requirement is ensured for network sizes up to 100 nodes for all the considered scenarios, including Scen 3 and Scen 4, where the MCR values are 6 and 8 min, respectively. This is justified via the DTR value of the second application, which is 4 min. There is, thus, no difference in the number of the second application’s messages every 1 or 3 min. Scen 1 and Scen 2 did not, therefore, present better delay values than Scen 3 and Scen 4. Similarly to the case of the delivery rate, the fourth application presents the worst delay values, since it is accommodated in a single timeslot and without any priority.

Figure 11 shows the control overhead for up to four applications. For all network sizes and scenarios, regardless of the number of applications, the MCR value is inversely proportional to the control overhead. Scen 2, which has the lower MCR value (1 min), presents the highest control overhead values. For three applications running and for a network size of 144 nodes, the control overhead values presented were 23,736.2 messages, 27,736.1 messages, 18,796.6 messages, and 17,296.4 messages for Scen 1, Scen 2, Scen 3, and Scen 4, respectively. This occurs since the lower MCR values mean more exchanged control messages.

Concerning energy consumption, Figure 12 shows that for all cases, the energy consumption increases with the number of applications. The lower values of MCR led to more energy consumption in most cases. In the case of four applications, as in Figure 12d and for a network size of 64 nodes, the presented values were 65,281.3 mj, 68,423.7 mj, 64,079.1 mj, and 62,819.8 mj for Scen 1, Scen 2, Scen 3, and Scen 4, respectively. This occurs since lower MCR values mean more activities concerning the control messages.

Figure 13 shows that for all cases, the control delivery rate is inversely proportional to the number of nodes. Scen 4, with the highest MCR value, presents the best control delivery rate values, whereas Scen 2, with the lowest MCR value, presents the worst ones. In the case of two applications (Figure 13b) and for a network size of 196 nodes, the control delivery rate values were 88.14% and 81.84% for Scen 4 and Scen 2, respectively. This is justified by the additional exchanged control messages for higher MCR values, which increases the probability of packet drop as a result of the buffer fullness.

Similarly to the control delivery rate, Figure 14 indicates that higher MCR values resulted in lower control delay values for all cases. For a network size of 100 nodes and four running applications, as depicted in Figure 14d, the control delay values were 1.57 s, 1.65 s, 1.42 s, and 1.29 s for Scen 1, Scen 2, Scen 3, and Scen 4, respectively. This occurs since for lower MCR values, the messages wait for more time in the buffer to be processed.

#### 6.2.2. Data Traffic Rate (DTR)

In this section, we evaluate the impact of the DTR on the network performance. We consider three different scenarios (Scen 1, Scen 5, and Scen 6) with different DTR values, as Table 4 shows.

Figure 15 depicts the data delivery rate for up to four applications running simultaneously. For a single application (Figure 15a), it is clear that the high DTR values significantly reduced the data delivery rate. Thus, Scen 6, which has the highest DTR value, presented the worst values. This occurs since the higher data traffic rate leads to more congestion and increases the probability of packet drop due to the buffer fullness. For Scen 1 and Scen 5, the AA approach ensured the data-delivery rate requirement for network sizes up to 225 nodes, compared with network sizes up to only 36 nodes in the case of Scen 6. To justify these findings, it should be noted that for a network size of 100 nodes, for instance, 98 data messages are sent per minute in the case of Scen 1 and Scen 5, compared with 5880 data messages per minute for Scen 6. For two applications, Figure 15b shows that Scen 1, with the lowest DTR value, continues to present better values than Scen 5 and Scen 6 for all cases. For the first and second applications, the satisfied network sizes are reduced from 196 nodes for Scen 1 to 100 nodes and 36 nodes for Scen 5 and Scen 6, respectively. Concerning the three applications, Figure 15c depicts that for the third application, which has a delivery rate requirement of 90%, the sizes of the satisfied networks increased from 36 nodes in the case of Scen 6 to 64 nodes and 225 nodes for Scen 5 and Scen 1, respectively. For four running applications, as Figure 15d shows, Scen 6, the scenario with the highest DTR value, continues to present the worst data-delivery rate values compared with Scen 1 and Scen 5. This is also applied to the fourth application, which has no priority. For a network size of 225 nodes and the fourth application, the data-delivery rate was 51.72% for Scen 1, compared with 44.53% and 26.81% for Scen 5 and Scen 6, respectively.

Concerning data delay, Figure 16 depicts that for a single application (Figure 16a), Scen 6 presents the highest delay values since it has the highest DTR value. Scen 1 and Scen 5 present similar values for the different network sizes. This occurs because there is just one application, and the DTR values for both Scen 1 and Scen 5 are hence equal. The delay requirement keeps ensuring network sizes up to 144 nodes for Scen 1 and Scen 5, compared with 64 nodes for Scen 6. For the case of two applications and the first application, as Figure 16b shows, the AA approach ensures the delay requirement for network sizes up to 100 nodes for Scen 1, compared with 144 nodes for the single application case. This could be justified using the second application, which raised the network traffic and prevented ensuring the delay requirement for network sizes larger than 100 nodes. The second application has a delay requirement of 950 ms, and the AA approach ensured the requirement for up to 100 nodes, 64 nodes, and 36 nodes for Scen 1, Scen 5, and Scen 6, respectively.

For three and four running applications (Figure 16c,d), the DTR value is directly proportional to the obtained delay values for all cases. This means that the scenarios with higher DTR values presented higher delay values. Concerning the fourth application, it presents the worst delay values since it is accommodated using a single timeslot and has no priority. For a network size of 196 nodes, the obtained delay values were 10.37 s, 16.37 s, and 24.37 s for Scen 1, Scen 5, and Scen 6, respectively. These values confirm, again, the high effect of the selected DTR value on the delay requirement.

Figure 17 shows that the control overhead increases with the number of nodes, regardless of the DTR value and the number of applications. For two and four applications, as Figure 17b,d show, all scenarios presented similar values for the network sizes up to 100 nodes. For larger network sizes, the control overhead is directly proportional to the DTR value. This means that Scen 6 with the highest DTR had the highest control overhead values, whereas Scen 1 had the lowest. For a network size of 196 nodes and four running applications, the control overhead values were 36,926.2 messages, 40,301.4 messages, and 47,097.8 messages for Scen 1, Scen 5, and Scen 6, respectively. This occurs since higher DTR requires more exchanged control messages (rescheduling request message, rescheduling message, and new scheduling message), as an attempt to ensure the application’s QoS requirements.

Figure 18 shows that, in general, the DTR value did not significantly affect the energy consumption. However, higher DTR values led to a little increase in energy consumption in most cases. For instance, in the case of three running applications and a network size of 100 nodes, the energy consumption values were 58,482.5 mj, 59,190.6 mj, and 60,647.2 mj for Scen 1, Scen 5, and Scen 6, respectively.

Figure 19 shows that Scen 1 and Scen 5 have roughly similar control delivery rate values. Scen 6, however, presents the worst values for all cases. This confirms that the control delivery rate is inversely proportional to the DTR value. For four applications and a network size of 144 nodes, Figure 19d shows that the obtained values were 59.64% for Scen 6, compared with 79.16% and 75.34% for Scen 1 and Scen 5, respectively. This is due to the additional control traffic in the case of Scen 6, which increases the congestion during the control timeslots after the network convergence.

The control delay is shown in Figure 20. Similarly to the control delivery rate, the obtained control delay values show that Scen 6 continues to present the worst values, whereas Scen 1 and Scen 5 present similar values for most cases. For two applications and a network size of 144 nodes, Figure 20b shows that the control delay values were 1.75 s for Scen 6, compared to 1.39 s and 1.47 s for Scen 1 and Scen 5 respectively.

Table 6 summarizes the previous results to provide an overview of the AA approach’s performance in terms of the data-delivery rate and data delay. All the data of Table 6 have adopted the obtained results for Scen 1 (with the default values of MCR and DTR, as Table 3 shows) and considered the first three applications (1st App, 2nd App, and 3rd App), which have QoS requirements.

## 7. Conclusions

This paper presented the application-aware (AA) approach, which adopted the IT-SDN framework on top of the TSCH technology. The proposed approach dynamically changes the current scheduling to adapt to the application’s QoS requirements. The AA approach isolates the different traffic types and supports the network scalability using shared timeslots. We employed two different strategies to evaluate the AA approach: (i) comparing it with the ATI approach, which assigns a single timeslot per application, and (ii) comparing the obtained results with the application’s QoS requirements in terms of delivery rate and delay. We considered up to four applications and network sizes of up to 225 nodes. Furthermore, we adopted six different scenarios to evaluate the impact of both MCR and DTR parameters on the network performance. The evaluation process took into account both the control and data planes, confirming the effectiveness of the AA approach. In comparison with the ATI approach, the AA approach increased the delivery rate by up to 28% and decreased the delay by up to 57%. Furthermore, our approach was capable of ensuring the application’s QoS requirements for the different network sizes. In future work, we plan to investigate the impact of other parameters on network performance, such as the difference rate and the application’s requirements. 

## Figures and Tables

**Figure 1 sensors-23-07143-f001:**
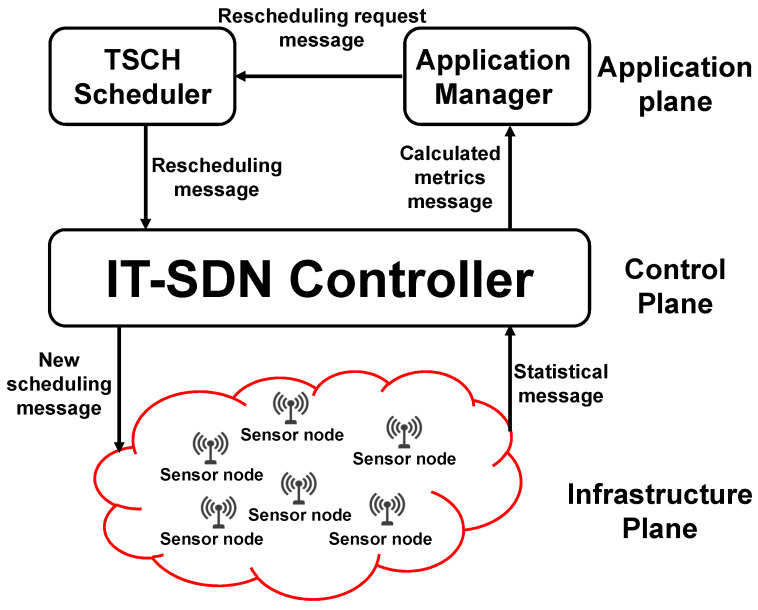
Application-aware (AA) IT-SDN system.

**Figure 2 sensors-23-07143-f002:**
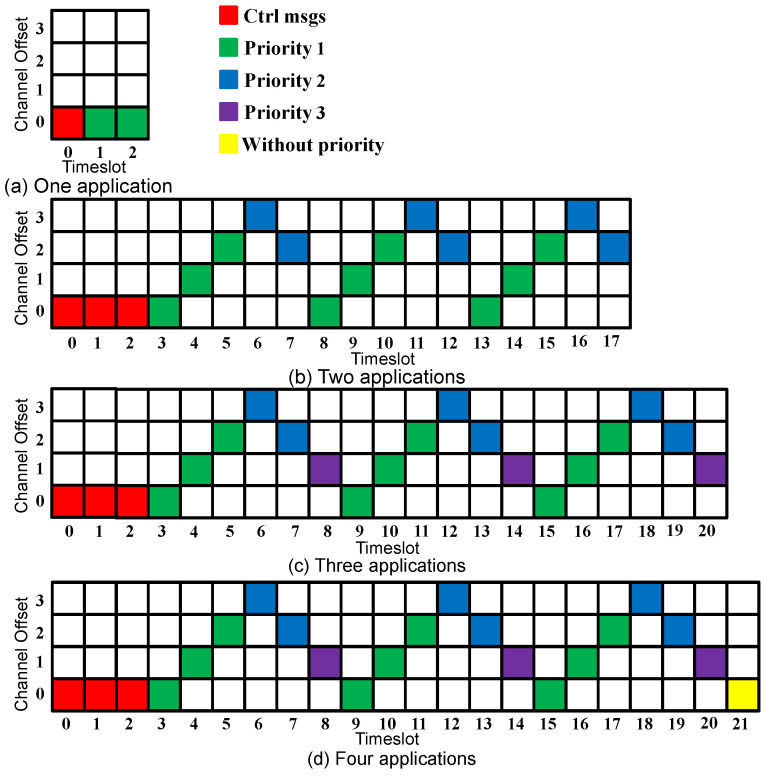
Sch 0 scheduling.

**Figure 3 sensors-23-07143-f003:**
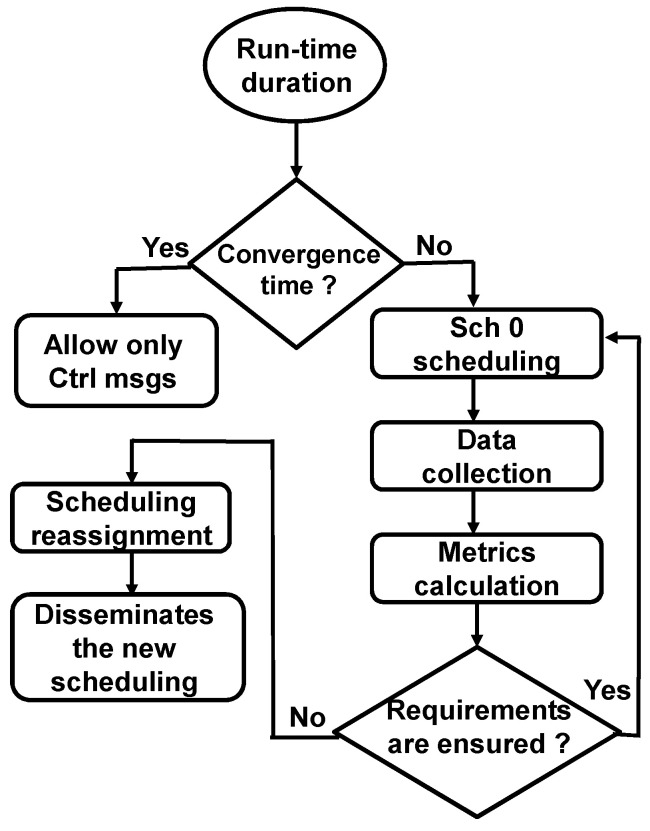
Sequence of the system operations.

**Figure 4 sensors-23-07143-f004:**
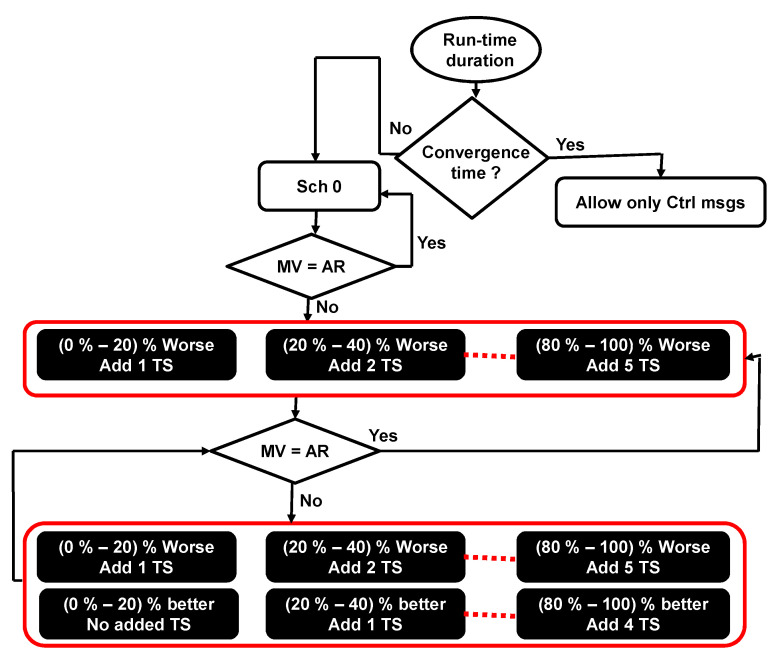
Scheduling calculation procedure.

**Figure 5 sensors-23-07143-f005:**
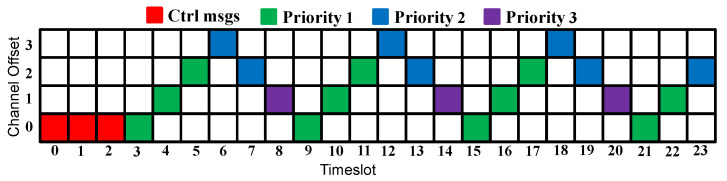
The new calculated scheduling (three-application case).

**Figure 6 sensors-23-07143-f006:**
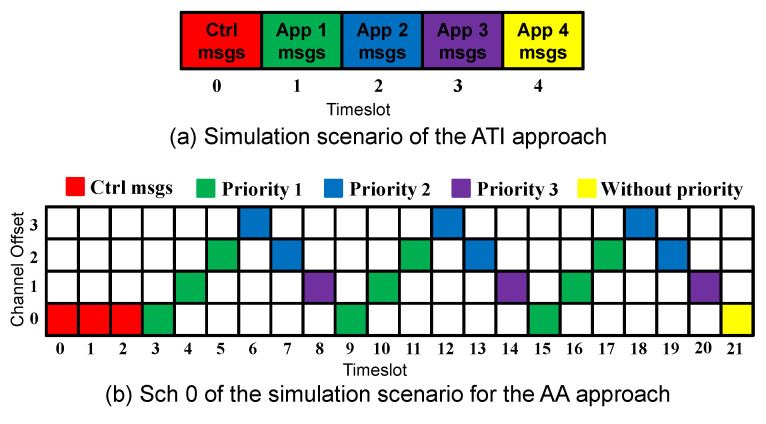
Simulation scenarios for ATI and AA approaches.

**Figure 7 sensors-23-07143-f007:**
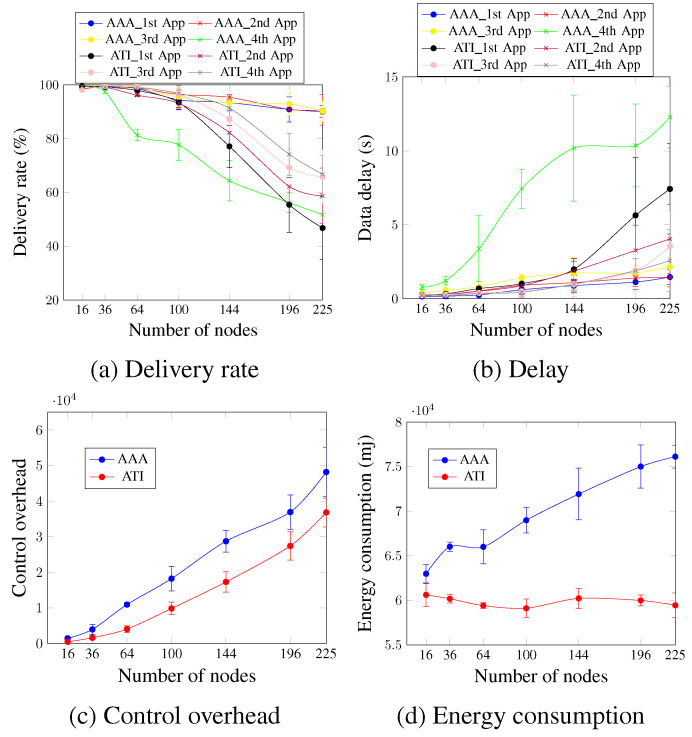
Comparison between AA and ATI approaches in the data plane (four applications’ case).

**Figure 8 sensors-23-07143-f008:**
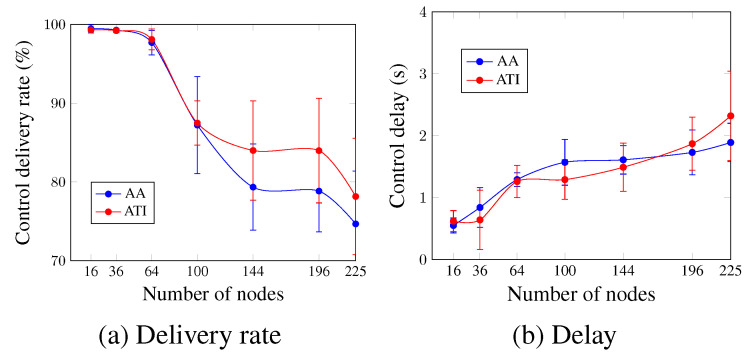
Comparison between AA and ATI approaches in the control plane (four applications’ case).

**Figure 9 sensors-23-07143-f009:**
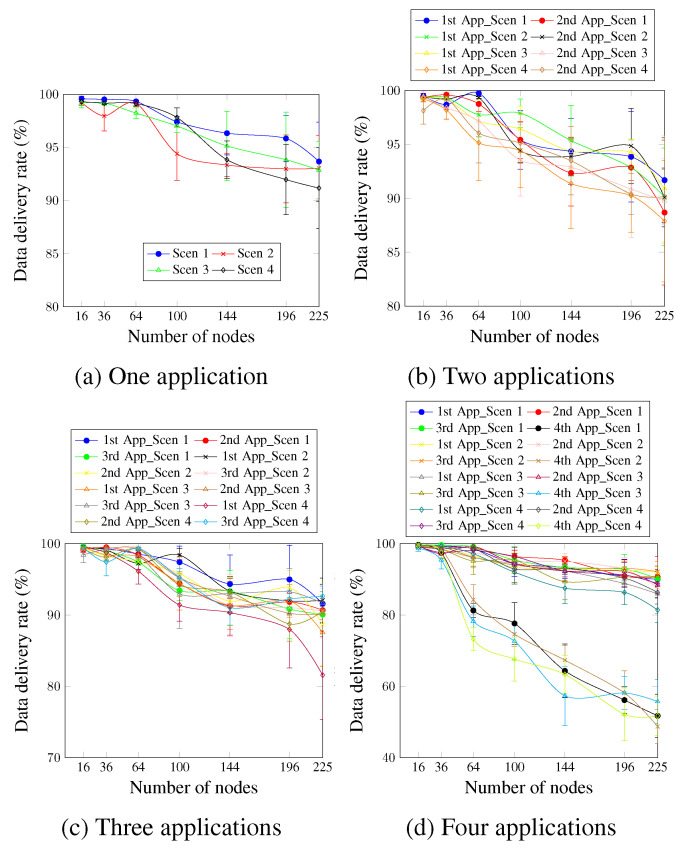
Data delivery rate, changing the MCR value.

**Figure 10 sensors-23-07143-f010:**
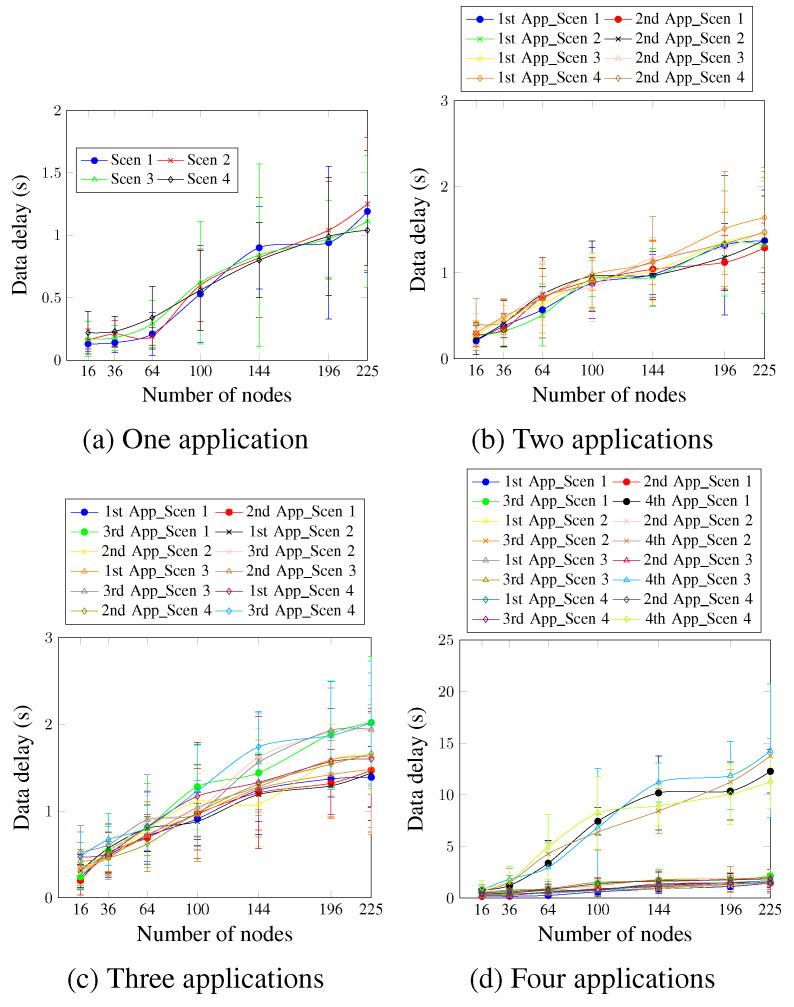
Data delay, changing the MCR value.

**Figure 11 sensors-23-07143-f011:**
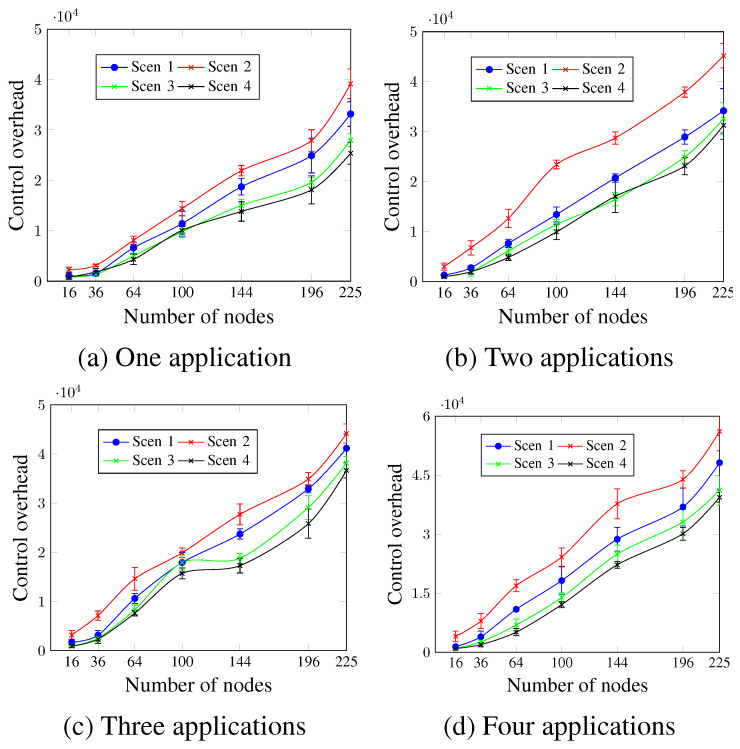
Control overhead, changing the MCR value.

**Figure 12 sensors-23-07143-f012:**
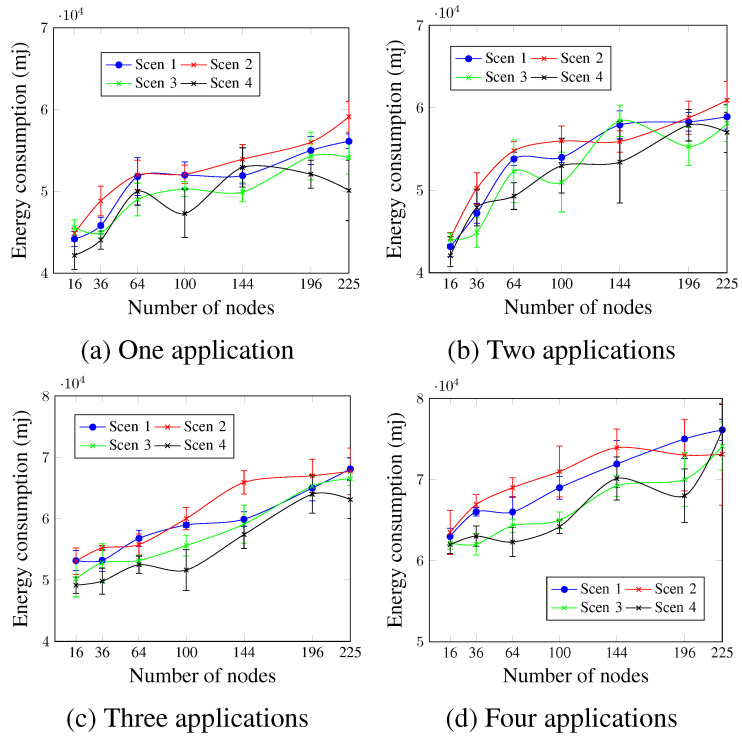
Energy consumption, changing the MCR value.

**Figure 13 sensors-23-07143-f013:**
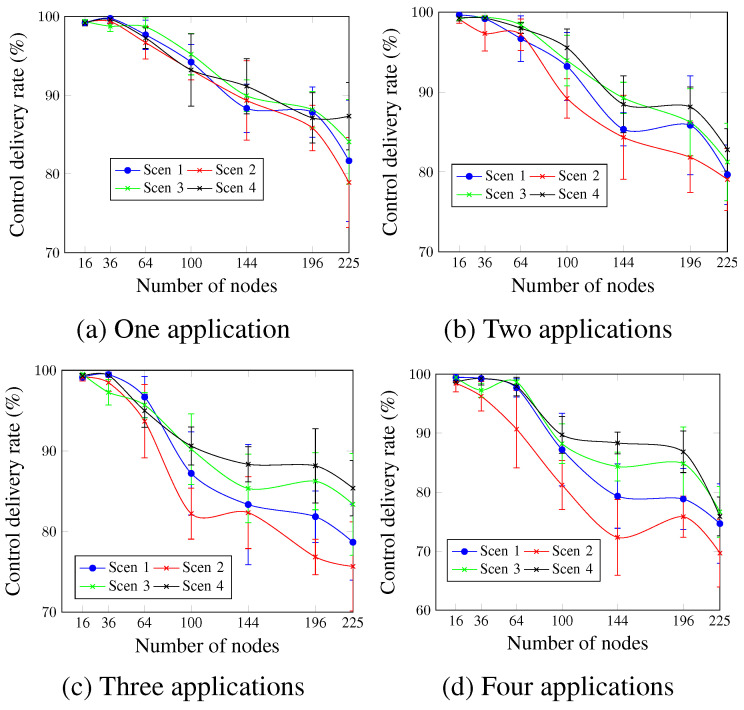
Control delivery rate, changing the MCR value.

**Figure 14 sensors-23-07143-f014:**
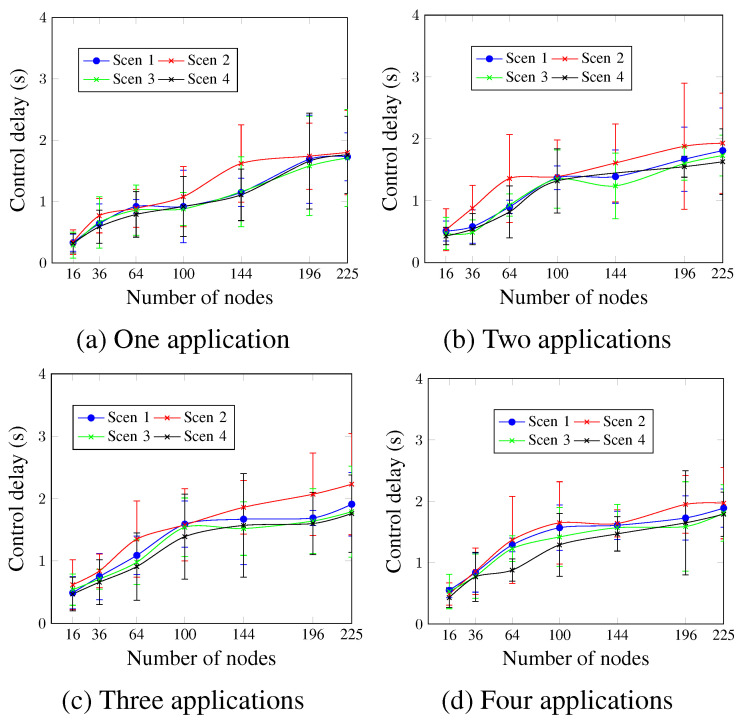
Control delay, changing the MCR value.

**Figure 15 sensors-23-07143-f015:**
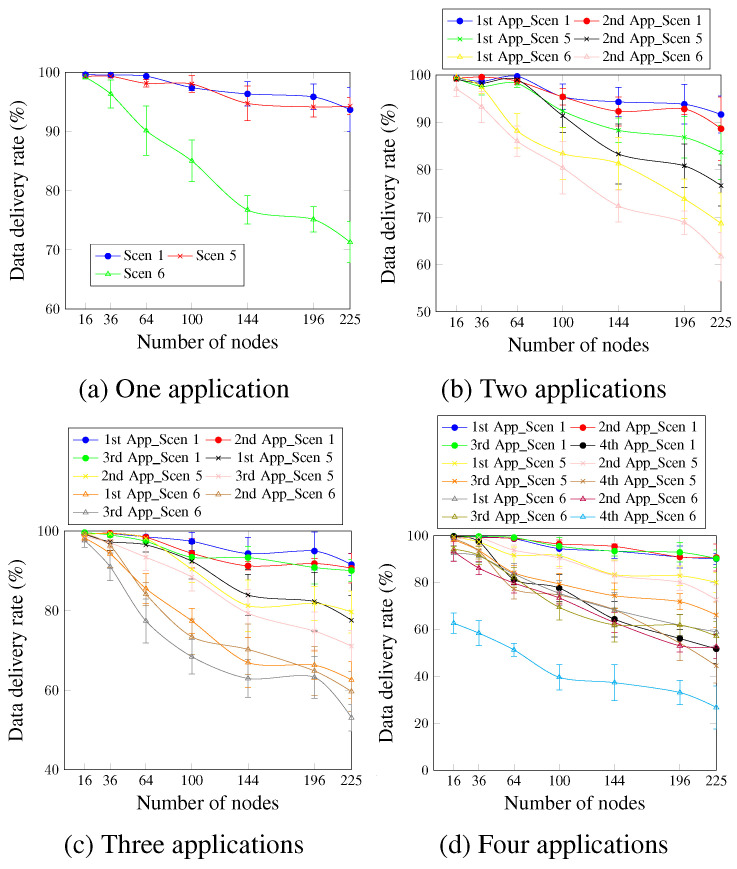
Data delivery rate, changing the DTR value.

**Figure 16 sensors-23-07143-f016:**
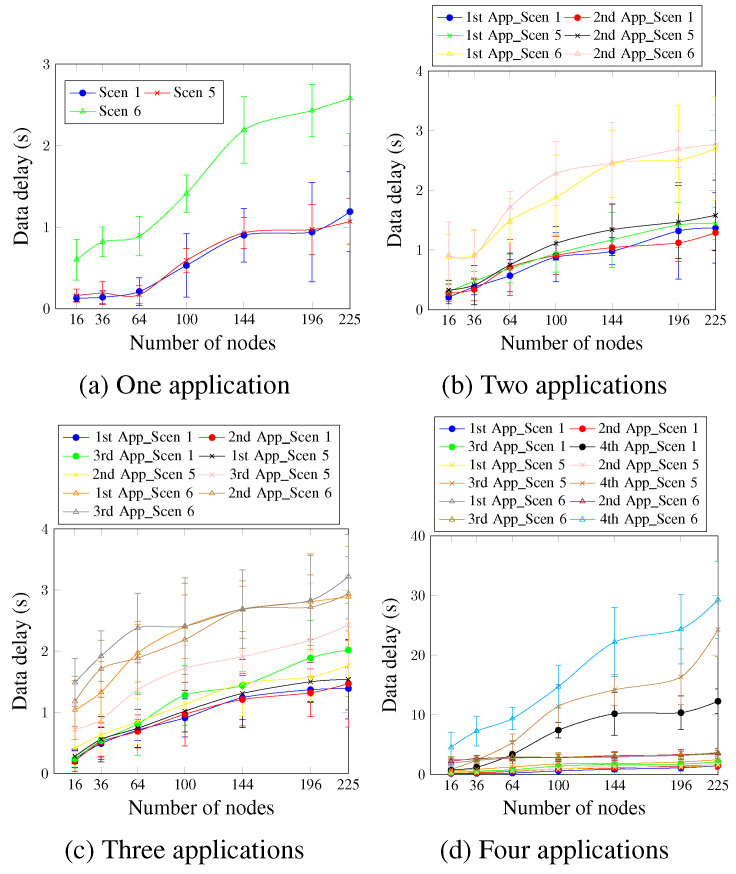
Data delay, changing the DTR value.

**Figure 17 sensors-23-07143-f017:**
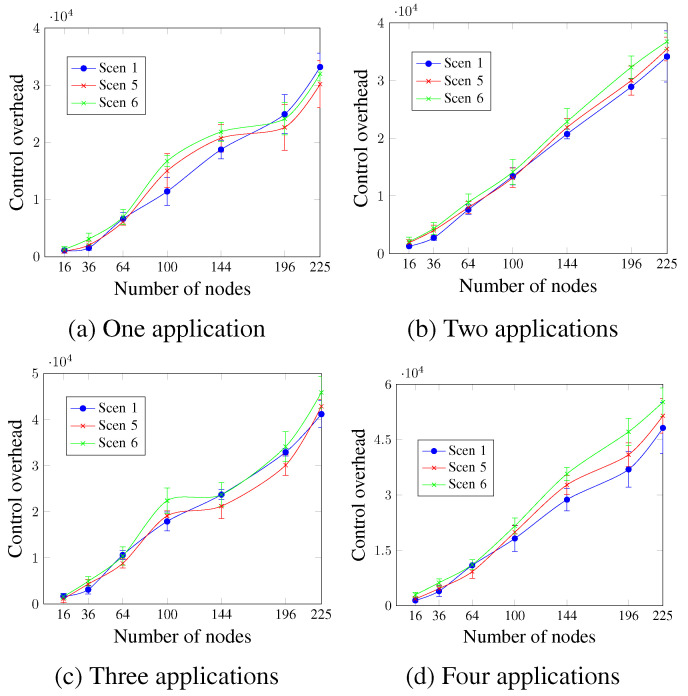
Control overhead, changing the DTR value.

**Figure 18 sensors-23-07143-f018:**
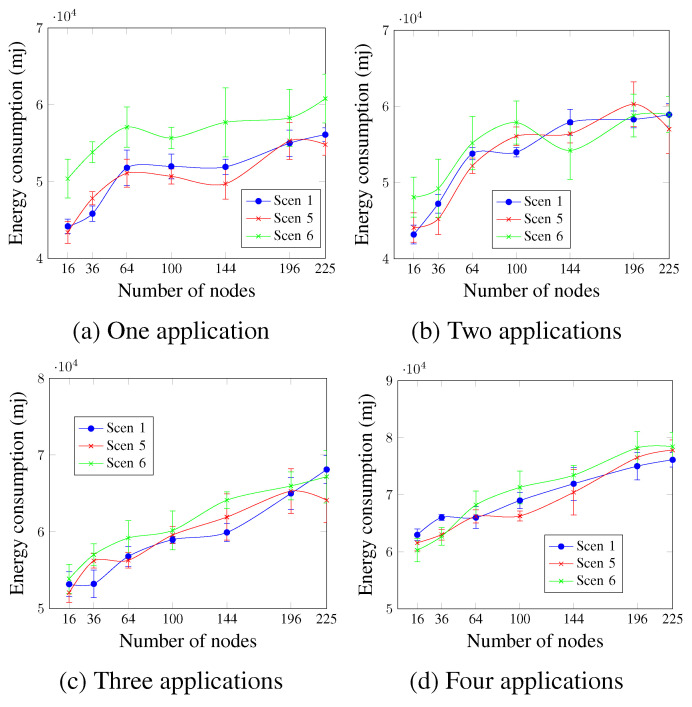
Energy consumption, changing the DTR value.

**Figure 19 sensors-23-07143-f019:**
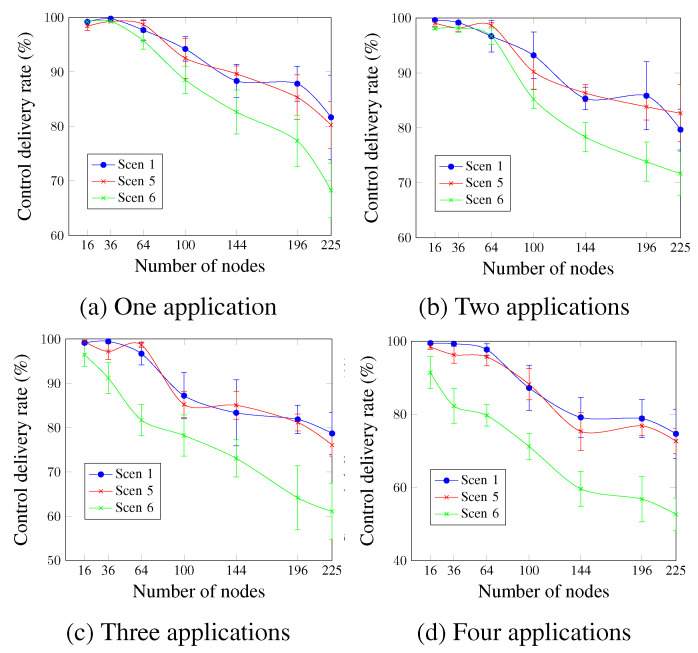
Control delivery rate, changing the DTR value.

**Figure 20 sensors-23-07143-f020:**
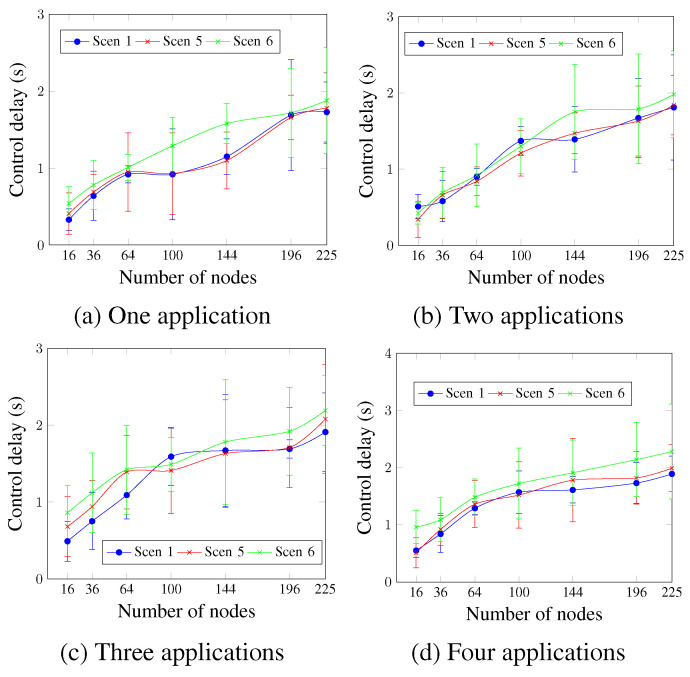
Control delay, changing the DTR value.

**Table 1 sensors-23-07143-t001:** Summary of the related work.

Work	Addressed Issue	Traffic Isolation Type	Timeslots Type	Scalability Support
Thubert et al. (2015) [15]	Interference and multipath fading	None	—	No
Baddeley et al. (2017) [14]	Control and data traffic competition	Control and data	Dedicated	No
Lo Bello et al. (2018) [17]	Mobility management	None	Dedicated	No
Orozco-Santos et al. (2021) [18]	Ensure the QoS level	Control and data/ applications’ traffics	Dedicated	No
Sayjari et al. (2021) [9]	Control and data traffic competition	Control and data	Shared	Yes
Orozco-Santos et al. (2022) [26]	Comparison among TSCH schedulers	—	—	—
Veisi et al. (2022) [19]	Control the application requirements	Control and data/ application’s traffics	Dedicated	No
Orozco-Santos et al. (2022) [20]	Scalability in the IWSNs	None	Dedicated	Yes
Sayjari et al. (2022) [16]	Application traffic competition	Control and data/ application’s traffics	Shared	Yes
This work	Ensure the application’s QoS requirements	Control and data/ applications traffics	Shared	Yes

**Table 2 sensors-23-07143-t002:** Priority levels of the considered application types.

	QoS Application’s Requirements		
Application Type	Delivery Rate	Delay	
Type 1	✓	✓	Priority 1
Type 2	—	✓	Priority 2
Type 3	✓	—	Priority 3
Type 4	—	—	Without priority

**Table 3 sensors-23-07143-t003:** Default simulation settings.

Topology	Square grid
Distance between neighbors	50 m
Compiling mote	Z1
Radio environment	UDGM
Simulation duration	3600 s
Simulation repetition	10 times for each case
Radio module power	0 dB
ContikiMAC channel check rate	16 Hz
IT-SDN version	0.4.1
Controller re-transmission timeout	2 s
ND protocol	Collect-based
CD protocol	None
Link metric	Expected Transmission Count (ETX)
Route recalculation threshold	20%
Size of the flow table	10 entries
Neighbor report max frequency	1 packet per minute
Route calculation algorithm	Dijkstra
Flow setup	Source routed
Data payload size	10 bytes
Number of TSCH channels	4 channels
Timeslot length	15 ms
Enhanced beacon (EB) transmission rate	2 s
Number of nodes	16,36,64,100,144,196,225
Number of applications (sinks)	1,2,3,4
Data traffic rate	1 packet per 1/4/8/10 min
Metric calculation rate	180 s
Difference rate	20%
Application’s requirements (1st App, 2nd App, 3rd App, 4th App)	
Delivery rate	92%, —, 90%, —
Delay	900 ms, 950 ms, —, —
Number of nodes, convergence time (s)	(16,73) (36,96) (64,117) (100,171) (144,196) (196,233) (225,329)

**Table 4 sensors-23-07143-t004:** Parameter’s values for the considered scenarios.

Scenario	MCR	DTR
Scen 1	180 s	DV
Scen 2	60 s	DV
Scen 3	300 s	DV
Scen 4	480 s	DV
Scen 5	DV	1/1/1/1 min
Scen 6	DV	1/4/8/10 s

**Table 5 sensors-23-07143-t005:** AA approach versus ATI approach.

Evaluation Metric	AA Vs ATI	
Data delivery rate	✓	AA outperformed
		Similar
		ATI outperformed
Data delay	✓	AA outperformed
		Similar
		ATI outperformed
Control overhead		AA outperformed
		Similar
	✓	ATI outperformed
Energy consumption		AA outperformed
		Similar
	✓	ATI outperformed
Control delivery rate		AA outperformed
	✓	Similar
		ATI outperformed
Control delay		AA outperformed
	✓	Similar
		ATI outperformed

**Table 6 sensors-23-07143-t006:** AA approach versus application’s QoS requirements.

Evaluation Metric	AA Approach Vs Application’s QoS Requirements	Number of Applications	
Data delivery rate	Ensured of up to 225 nodes	One App	Performance of the 1st App
	Ensured of up to 196 nodes	Two Apps	
	Ensured of up to 196 nodes	Three Apps	
	Ensured of up to 144 nodes	Four Apps	
Data delay	Ensured of up to 144 nodes	One App	
	Ensured of up to 100 nodes	Two Apps	
	Ensured of up to 100 nodes	Three Apps	
	Ensured of up to 144 nodes	Four Apps	
Data delivery rate	—	Two Apps	Performance of the 2nd App
	—	Three Apps	
	—	Four Apps	
Data delay	Ensured of up to 100 nodes	Two Apps	
	Ensured of up to 64 nodes	Three Apps	
	Ensured of up to 100 nodes	Four Apps	
Data delivery rate	Ensured of up to 225 nodes	Three Apps	Performance of the 3rd App
	Ensured of up to 225 nodes	Four Apps	
Data delay	—	Three Apps	
	—	Four Apps	

## Data Availability

Not applicable.

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
