# Peer review of "Application-Aware Scheduling for IEEE 802.15.4e Time-Slotted Channel Hopping Using Software-Defined Wireless Sensor Network Slicing"

_sensors, 2023, doi:10.3390/s23167143_

Round 1

Reviewer 1 Report

The article proposes Application-aware scheduling for IEEE 802.15.4e TSCH using SDWSN slicing. The article cannot be accepted in its current form. However, I propose to improve the article by addressing the following points:

1. More numerical/statistical results should be added to the abstract.

2. The abstract needs to be improved by highlighting the study's novelty.

3. The introduction chapter should be extended with more recent references.

4. Figure 2,5 and 6 – columns and rows labelling are missing.

5. I do not find any statistical analysis of the results. The author should calculate the p-value to statistically validate the results.

6. Authors need novelty and originality in their work. They need to establish the clear superiority of their proposed methodology through comprehensive comparison results with very recent methods.

7. The article appears to be a technical report on scheduling for IEEE 802.15.4e TSCH using SDWSN slicing The authors need to compare their proposed model with existing models.

8. In the proposed model, there is a lack of detailed explanation as to why better results are obtained.

9. I did not find any comprehensive comparative analyses. More technical comparisons with other existing methods should be provided.

10. It is suggested to empower the discussion section by highlighting the novelty and numerical/statistical findings of this study.

11. The number of reference papers cited is not enough. There are more reference papers to review regarding recent developments. There are at least 15 to 20 papers published during the past five years that must be referred to in a research paper.

Reviewer 2 Report

I see the authors took out time to do a good job. However there are my concerns and it need to be addressed.

The statement "The evaluation process considered up to four applications with different 10 QoS requirements in terms of delivery rate and delay" the author did not mention these four application?

To the best of our knowledge, this approach is the first  to support network scalability using shared timeslots, ensuring at the same time the application’s QoS level. Are you sure. I am not. I will advice you play low key on this so that reviewers don't take the author up for such claims.

I believe the introduction and related work are ok but  could be improve by looking at some other detailed study/literature. this will in-turn make the two sections more robust. for instance,

“Efficient Topology Discovery Protocol for Software Defined Wireless Sensor Network” Bulletin of Electrical Engineering and Informatics, Vol. 11, No. 1, February 2022, pp. 256~269, ISSN: 2302-9285, DOI: 10.11591/eei.v11i1.3240.

          1.         Integrating Artificial Intelligence (A.I), Internet of Things (IoT) and 5G for Next-Generation Smart grid: A Survey of Trends and Prospect” IEEE Access, Vol 10, pp. 4794-4831, 2022 .

 Design and Implementation of Intrusion Detection Systems using RPL and AOVD Protocols-based Wireless Sensor Networks” International Journal of Electronics and Telecommunications, 2023, vol. 69, no. 2, pp. 309-318, DOI: 10.24425/ijet.2023.144366.

and many more.

The results are too small. they could be made a bit bigger for clarity/readability.

The References are not in order. in fact the author need to take out time to write following the template of sensor mdpi. For example look at reference 1 and 9....they do not match in style.

So far the work is good. Let the authors just adhere to the comments and youre good to go.

Accept with minor corrections

I see the authors took out time to do a good job. However there are my concerns and it need to be addressed.

The statement "The evaluation process considered up to four applications with different 10 QoS requirements in terms of delivery rate and delay" the author did not mention these four application?

To the best of our knowledge, this approach is the first  to support network scalability using shared timeslots, ensuring at the same time the application’s QoS level. Are you sure. I am not. I will advice you play low key on this so that reviewers don't take the author up for such claims.

I believe the introduction and related work are ok but  could be improve by looking at some other detailed study/literature. this will in-turn make the two sections more robust. for instance,

“Efficient Topology Discovery Protocol for Software Defined Wireless Sensor Network” Bulletin of Electrical Engineering and Informatics, Vol. 11, No. 1, February 2022, pp. 256~269, ISSN: 2302-9285, DOI: 10.11591/eei.v11i1.3240.

          1.         Integrating Artificial Intelligence (A.I), Internet of Things (IoT) and 5G for Next-Generation Smart grid: A Survey of Trends and Prospect” IEEE Access, Vol 10, pp. 4794-4831, 2022 .

 Design and Implementation of Intrusion Detection Systems using RPL and AOVD Protocols-based Wireless Sensor Networks” International Journal of Electronics and Telecommunications, 2023, vol. 69, no. 2, pp. 309-318, DOI: 10.24425/ijet.2023.144366.

and many more.

The results are too small. they could be made a bit bigger for clarity/readability.

The References are not in order. in fact the author need to take out time to write following the template of sensor mdpi. For example look at reference 1 and 9....they do not match in style.

So far the work is good. Let the authors just adhere to the comments and youre good to go.

I will like to review this work again if need arise.

Accept with minor corrections

Reviewer 3 Report

- I recommend using English proofreading to check English (f.e. MDPI)

- Section 3.2 please add some text

- I recommend accepting article

- I recommend using English proofreading to check English (f.e. MDPI)
